# Depth of acid penetration and enamel surface roughness associated with different methods of interproximal enamel reduction

**Gholamreza Danesh**[1]☯*, **Pascal Kai Konstantin Podstawa**[1]☯, **Cate-Emilia Schwartz**[1]☯, **Christian Kirschneck**[2]☯, **Mozhgan Bizhang**[3]☯, **Wolfgang H. Arnold**[4]☯

1 Department of Orthodontics, Faculty of Health, School of Dentistry, Witten/Herdecke University, Witten, Germany, 2 Department of Orthodontics, University Medical Centre of Regensburg, Regensburg, Germany, 3 Department of Operative and Preventive Dentistry, Faculty of Health, School of Dentistry, Witten/Herdecke University, Witten, Germany, 4 Department of Biological and Material Sciences in Dentistry, Faculty of Health, School of Dentistry, Witten/Herdecke University, Witten, Germany

☯ These authors contributed equally to this work.
* gholamreza.danesh@uni-wh.de

**Data Availability Statement:** All relevant data are within the manuscript and its Supporting Information files.

## Abstract

### Objectives

To assess and compare the enamel surface quality after interproximal enamel reduction (IPR) was performed with different systems and to study the relation between acid penetration depth and enamel surface quality as well as the importance of remineralization.

### Methods

Sixty-five extracted teeth were randomly allocated to five experimental groups: untreated control, manual with New Metal Strips, mechanical with oscillating segment (OS) discs, Safe-Tipped Bur Kit, and the Ortho-Strip, followed by 30 s of polishing with the Softflex system and the Compo-system after treating the tooth with OS discs. Mesial surfaces were demineralized for 24 h and distal surfaces were subjected to interchanging demineralization and remineralization cycles of 24 h each for 18 days. The analysis was carried out by profilometry, scanning electron microscopy, and polarization microscopy.

### Results

After IPR and polishing, enamel roughness was reduced for all systems tested except for the Essix Safe-Tipped Bur Kit. Subsequent demineralization increased enamel roughness in all groups except controls beyond the original level prior to IPR except for IPR with New Metal Strips or Ortho-Strips and subsequent polishing. Cyclic demineralization and remineralization for 18 days yielded a reduction in acid penetration depth and an increase in surface smoothness, which correlated with each other only for controls and treatment with New Metal Strips or Ortho-Strips.

**Funding:** The authors received no specific funding for this work

**Competing interests:** The authors have declared that no competing interests exist.

## Conclusions

Manual IPR, using New Metal Strips and, even more, the oscillating IPR system Ortho-Strips, yielded smoother interproximal enamel surfaces and less acid penetration depth than the IPR systems with OS discs and the Safe-Tipped Bur Kit after polishing and 18 days of cyclic demineralization and remineralization. Irrespective of the IPR procedure, proper remineralization of IPR-treated surfaces is advisable to reduce caries susceptibility.

## Introduction

In recent years, orthodontists have increasingly focused their interest on non-extraction therapy to resolve crowding and loss of space within the dental arch. Interproximal enamel reduction (IPR) is a frequently used alternative to extraction therapy to gain space and to treat moderate dental crowding [1].

Apart from the benefits of IPR in resolving dental crowding, it is still an invasive technique, which produces irreversible loss of hard tooth structures [2]. Different reports exist in the literature regarding the maximum amount of enamel reduction admissible without inducing permanent damage to the teeth and the oral system. Because of the reduced enamel thickness of teeth in the anterior mandibular region, the use of IPR on lower incisors is severely limited [3]. Thus it is generally recommended that clinicians assess enamel thickness radiographically before extensive IPR, since individual variation is known to exist [4].

Since the beginning of the clinical application of the IPR procedure, a potential increase in plaque accumulation and caries susceptibility of the abraded enamel surfaces after IPR has been postulated [5,6]. Increased surface roughness of enamel after performing IPR could promote plaque accumulation and enamel demineralization by acids produced by cariogenic bacteria in the forming biofilm [7,8]. The application of fluoride varnishes after IPR has been recommended to alleviate this problem [9], but has been shown to be sometimes ineffective, in particular, if application frequency is too low [10]. Thus the aim of our study was to assess and compare the enamel surface quality after IPR was performed with four different commercially available systems and to study the relationship between acid penetration depth and enamel surface quality as well as the importance of remineralization. If acid penetration depth is directly related to enamel surface quality, the IPR systems and techniques identified that produce the smoothest enamel surface could have a clinical advantage in this regard.

## Materials and methods

### Specimens and preparatory measures

The protocol for the collection of teeth for this in vitro study was approved by the ethics committee of Witten/Herdecke University, Witten, Germany (No. 116/2013). Sixty-five extracted, caries-free, and intact human lower front teeth were included in this study. The soft tissues and the calculus were removed by using a toothbrush, toothpaste, and, if required, a polishing handpiece, rubber cup/brush, fluoride-free polishing paste, and a scaler. First, a visual and radiological analysis for the presence of enamel decalcification and caries, as well as an examination of the soundness of the interproximal surfaces of each tooth, was conducted.

A digital X-ray unit (Heliodent DS Sidexis, D3495 XIOX Plus Wall Module, Sirona, Bensheim, Germany) with a large X-ray sensor fixed within a silicone mold (Alphsil Putty Soft,

**Table 1. Listing of tested IPR systems.**

|   | IPR system | code | Manufacturer | Procedure/ handpiece | Manufacturer |
|---|------------|------|--------------|----------------------|--------------|
| 1 | New Metal Strips | GMS | GC, Tokyo, Japan | *Manual* | N.A. |
| 2 | OS-discs (ASR set) | KAS | Komet, Lemgo, Germany | *Mechanical*/Komet OS 30 | Komet, Lemgo, Germany |
| 3 | Ortho-Strip system | IOS | Intensiv Dental, Montagnola, Switzerland | *Mechanical*/Intensiv Swingle WG-69 LT | W&H, Laufen/Obb. Germany |
| 4 | Safe-Tipped Bur Kit | STB | Raintree Essix, LA, USA | *Mechanical*/Kavo, Intracompact 25LHC | Kavo, Biberbach, Germany |

Müller-Omicron-Dental, Germany) was used. The extracted and caries-free teeth were stored throughout in physiological NaCl solution containing 0.1% thymol.

## Interproximal enamel reduction

The 65 teeth were randomly distributed to five experimental groups of 13 specimens each: a reference group not treated by IPR as control and four experimental IPR groups. Table 1 shows three mechanical systems [Komet ASR-Set (KAS), Safe-Tipped Bur (STB) Kit, Intensiv Ortho-Strip-System (IOS)] and one manual system, the GC New Metal Strips (GMS), which were used for IPR and compared in this study. To reproduce the physiological movement of teeth during the grinding, the extracted teeth were mounted using silicon material (Optosil; Heraeus Kulzer, Hanau, Germany), and interproximal contacts were reestablished. The IPR was conducted according to the respective manufacturers' recommendations under water-cooling. IPR of 0.2 mm was performed at both interproximal surfaces of each tooth, followed by re-contouring to produce an adequate interproximal shape and polishing with the Softflex system (3M, St. Paul, MN) and the Komet ASR-Set (KAS) CompoClips for 30 seconds to mitigate roughness. If necessary, teeth were separated by using interdental wedges.

## Demineralization and remineralization procedure

Interproximal areas were checked under a light microscope (Leica WILD M3Z, Wetzlar, Germany) and the remaining enamel surface was covered in wax. Mesial interproximal tooth surfaces were designated for only 24 hours (h) of demineralization at 37˚C, whereas distal tooth surfaces underwent demineralization and remineralization cycles interchanging every 24 hours for 18 days at 37˚C. This was achieved by inserting the teeth into a respective demineralization (pH 4.67–4.73) or remineralization solution (pH 6.9–7.0). The composition of the demineralization solution was: 1.6 g (1.6%) hydroxyethyl cellulose (HEC), 33 ml distilled water, 15 ml KCl (1 mol/l), 33 ml sodium acetate (0.2 mol/l), 15 ml acetic acid (0.2 mol/l), 1 ml $KH_2PO_4$ (90 mmol/l), and 1 ml $CaCl_2$ (150 mmol/l). The remineralization solution comprised 83 ml of distilled water, 15 ml KCl (1 mol/l), 1 ml $KH_2PO_4$ (90 mmol/l), and 1 ml $CaCl_2$ (150 mmol).

The experiment started with the demineralization process for 24 h, after which the mesial tooth surfaces were covered in wax to protect them from further demineralization and remineralization. Teeth were rinsed during each interim and solution change with bidistilled water to avoid contamination between solutions. Temperature and pH value were checked on a regular basis.

## Profilometry

Profilometry was used to assess surface roughness at each interproximal area of interest before and after IPR, after demineralization for 24 h, and after the demineralization/remineralization cycles lasting for 18 days. All teeth were analyzed by an optical profilometer (InfiniteFocus, G3, Alicona Imaging GmbH, Graz, Austria). For profilometry analysis, teeth were temporarily

demounted from their silicon mold and then reinserted in their original position for the continuation of the experiments. Image processing was realized with the corresponding software (Alicona Imaging GmbH, Graz, Austria). The surfaces were magnified 50 times in the area of the examination window and analyzed by line measurements (Ra) and area measurements (asfc). The computer analysis of the surface characteristics permitted a numeric and graphical description of each tooth surface, quantifying enamel roughness in average μm on each surface in the range of a section of the examination windows (Fig 1).

## Scanning electron microscopy

Scanning electron microscopy (SEM) (Zeiss Sigma VP, Carl Zeiss, Oberkochen, Germany) at zoom factors of 500, 750, and 1.000 was used to compare interproximal enamel surface roughness of randomly chosen teeth, visually and qualitatively, of each group before and after IPR, after demineralization for 24 h, and after the demineralization/remineralization cycles lasting for 18 days, and to corroborate the quantitative profilometry results. The specimens were prepared according to standard operating procedures and sputtered with gold-palladium. SEM analysis was carried out at 20 kV, using a secondary electron detector.

## Polarization microscopy

Before histological examination by polarization microscopy, the teeth were prepared by separating the roots with a diamond disc (Horico, Berlin, Germany) and the crowns were embedded in methyl methacrylate (MMA)-based resin (Technovit, Heraeus Kulzer, Hanau, Germany). They then went through a dewatering process comprising 24 h of 70% isopropanol, 24 h 90% isopropanol, 2 x 24 h isopropanol, and 3x daily 5–10 min vacuum, 2 x 48 h xylene, and 3x daily 5–10 min vacuum and, finally, four days in infiltration solution and 3x daily 5–10 min vacuum. Samples were then fixed on a 7 mm-thick base with superglue. The polymerization of Technovit in the powder (polymer) liquid (monomer) technique was carried out as

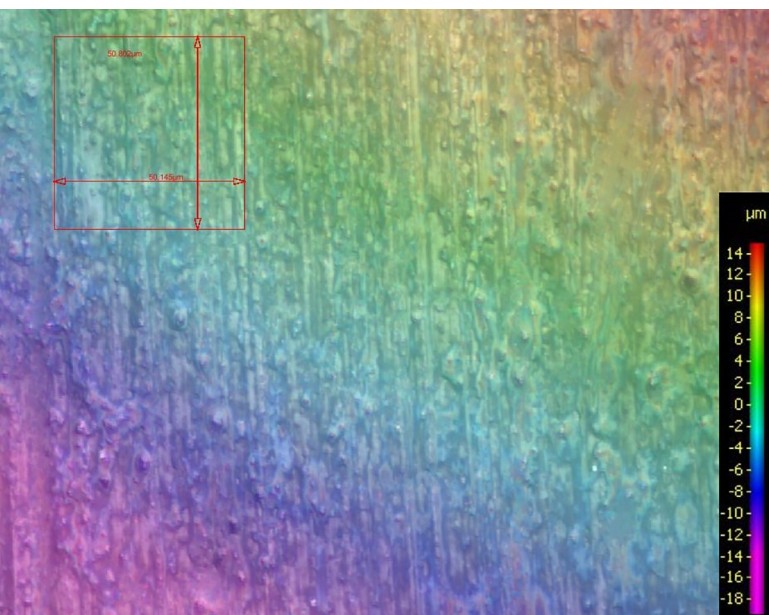

**Fig 1. Profilometry after IPR with the Ortho-Strip system.** The legend shows the position of the specimen according to a horizontal layer. The quadrangle shows the measured area.

follows: 5 min light vacuum with the water jet pump, 10 min at 4˚C in a desiccator, 4 days at –17˚C under oxygen exclusion, 4 h at 4˚C, and 1 day at room temperature. With a sawmill microtome (Leica microtom SP 1600, Wetzlar, Germany), three grinding samples per tooth (80–90 μm) were cut off from the center after checking the appropriate hardness of the samples and then visualized under the polarization microscope (Leica, Wetzlar, Germany) and evaluated with the appropriate software application suite.

Depth of acid penetration (μm) was measured quantitatively at 10 measuring points at both the mesial and the distal surface of each grinding sample, with the Leica software (Leica Application Suite, version 4.0.0); spotted from two sections; and photographically documented. Thus, 60 measurements per tooth (30 mesial/30 distal) were documented.

## Statistical analysis

Statistical analysis was performed using the IBM® SPSS® Statistics 23 (IBM, Armonk, NY, USA) application. A descriptive-exploratory data analysis was performed to assess the assumptions of parametric testing (normality, homogeneity of variance). The arithmetic mean (M) with standard deviation (SD) and 95% confidence interval (CI) of the mean were calculated as descriptive statistics. The respective changes of enamel roughness directly after IPR, after another 24 h of demineralization or 18 days of cyclic demineralization and remineralization as well as acid penetration depth, were compared between groups by Kruskal-Wallis H tests. One-sample $t$-tests versus 0 (bootstrapped, bias-corrected, and accelerated, 10.000 samples) were used to determine the significance of enamel roughness changes. Possible correlations between enamel surface roughness and acid penetration depth were determined by means of a Spearman correlation. Significance was assumed at $p \leq 0.05$.

## Results

### Profilometry

Table 2 shows the results of the profilometry analysis after IPR and polishing. The control group was used as a reference. The various IPR systems differed significantly in terms of their effect on enamel surface roughness after polishing (p = 0.004). GMS (p = 0.008), KAS (p = 0.01), and IOS (p = 0.019) each resulted in a significant reduction in enamel surface

**Table 2. Enamel surface roughness at the beginning (after tooth-cleaning) and after IPR (n = 26 per group, pooled mesial and distal tooth surfaces of the 13 teeth per group).**

| Enamel surface roughness [μm] | Mesial and distal tooth surfaces (n = 26 per group) | | |
|---|---|---|---|
| | After tooth-cleaning (start) M±SD [95 % CI] | Change Δ by IPR+polishing M±SD [95% CI] p (significance Δ >0) | After IPR+polishing M±SD [95% CI] |
| New Metal Strips (GMS) | 27.3 ±16.7 [20.6/34.0] | -12.6 ±17.8 [-19.8/-5.4] **p = 0.008**\* | 14.8 ±10.4 [10.6/1.0] |
| OS-discs (KAS) | 21.0 ±12.3 [16.0/25.9] | -6.5 ±11.3 [-11.1/-1.9] **p = 0.01**\*\* | 14.4 ±6.0 [12.0/16.9] |
| Ortho-Strips (IOS) | 33.8 ±19.5 [25.9/4.,7] | -8.9 ±18.0 [-16.1/-1.6] **p = 0.019**\* | 24.9 ±15.6 [18.6/31.2] |
| Safe-Tipped Bur Kit (STB) | 23.5 ±12.9 [18.3/28.7] | +10.5 ±29.9 [-1.6/22.6] p = 0.109 | 34.0 ±24.3 [24.2/43.8] |
| Control group | 28.0 ±16.4 [20.7/35.3] | 0.0 ±0.0 [0.0/0.0] p = 1.000 | 28.0 ±16.4 [20.7/35.3] |
| | | H(4) = 15.261 **p = 0.004**\*\* | |

M = arithmetic mean; SD = standard deviation; CI = confidence interval of the mean; p = statistical significance; p (Δ >0) one-sample $t$-test (bootstrapped, bias-corrected, and accelerated, 10.000 samples) to test significant differences of change Δ from 0; H (df) = test statistic (degrees of freedom) of Kruskal-Wallis H test, comparing changes Δ between groups

roughness after polishing (Table 2), whereas the STB (p = 0.109) caused a non-significant increase in surface roughness after polishing (no change in the control group without IPR).

**After demineralization/remineralization cycles.** After 24 h of demineralization, it was found that the surface roughness of enamel surfaces treated with KAS (p = 0.028) and IOS (p = 0.012) increased significantly compared to directly after IPR and polishing, whereas for GMS (p = 0.052) and STB (p = 0.225), this increase was not significant (Table 3). In the control group without IPR, demineralization induced a small, non-significant decrease in surface roughness. Overall, however, no significant differences between experimental groups were found regarding surface roughness after demineralization, compared to after IPR and polishing (p = 0.072). Compared to initial surface roughness, however, the effects of demineralization differed for the various IPR systems tested (p = 0.002, Table 3). KAS (p = 0.033) and STB (p = 0.001) produced significantly higher enamel roughness after demineralization, whereas approximately initial roughness values were reached for the other systems after demineralization (Table 3).

After 18 days of interchanging 24 h demineralization and remineralization cycles, significant differences in surface roughness were observed across all experimental groups (p = 0.002). GMS (p = 0.018) and KAS (p = 0.001) were associated with significantly increased enamel roughness compared to the situation directly after IPR and polishing, whereas this increase was present but not significant for the STB (p = 0.147, Table 3) group. In the control group without IPR and the Ortho-Strips group (IOS), a non-significant decrease in surface roughness compared to directly after IPR and polishing could be observed. Compared to initial surface roughness, the different IPR systems tested did not show significant differences in enamel roughness after 18 days of cyclic demineralization and remineralization (p = 0.106, Table 3).

### Scanning electron microscopy

The visual-qualitative assessment of enamel surface roughness based on SEM images of interproximal areas of randomly chosen teeth per group corroborated to a high degree with the results of the profilometry analysis (Fig 2).

**Table 3. Enamel surface roughness after 24 h of demineralization at 37°C (n = 13, mesial tooth surfaces) or following interchanging demineralization and remineralization cycles of 24 h for 18 days at 37°C (n = 13, distal tooth surfaces).**

| Enamel surface roughness [μm] | Mesial tooth surface (n = 13 per group, only 24 h of demineralization at 37°C) | | | Distal surface (n = 13 per group, interchanging de- and remineralization cycles of 24 h for 18 days at 37°C) | | |
|---|---|---|---|---|---|---|
| | Change Δ demin. (to IPR+polish.) M±SD [95% CI] p (Δ > 0) | After demin. (after IPR) M ±SD [95% CI] | Change Δ demin. (to start) M±SD [95% CI] p (Δ > 0) | Change Δ demin./remin. (to IPR+polish.) M±SD [95% CI] p (Δ > 0) | After demin./remin. (after IPR) M±SD [95% CI] | Change Δ demin./remin (to start) M±SD [95% CI] p (Δ > 0) |
| New Metal Strips (GMS) | +9.1 ±14.4 [0.0/18.3] p = 0.052 | 22.9 ±10.2 [16.4/29.4] | -0.6 ±17.4 [-11.6/10.5] p = 0.901 | +15.7 ±19.1 [3.6/27.8] **p = 0.018**\* | 31.5 ±14.8 [22.1/40.9] | -0.5 ±21.7 [-14.3/13.3] p = 0.942 |
| OS-discs (KAS) | +17.1 ±19.7 [4.6/29.6] **p = 0.028**\* | 30.6 ±17.1 [19.7/41.4] | +13.7 ±18.2 [2.1/25.3] **p = 0.033**\* | +14.2 ±9.0 [8.5/19.9] **p = 0.001**\*\*\* | 30.2 ±11.7 [22.7/37.6] | +4.2 ±18.4 [-7.5/15.9] p = 0.441 |
| Ortho-Strips (IOS) | +12.3 ±13.9 [3.5/21.1] **p = 0.012**\* | 28.0 ±11.8 [20.5/35.6] | +0.3 ±12.1 [-7.4/7.9] p = 0.949 | -10.0 ±20.3 [-22.9/3.0] p = 0.126 | 25.4 ±11.0 [18.4/32.4] | -15.1 ±27.2 [-32.4/2.1] p = 0.082 |
| Safe-Tipped Bur Kit (STB) | +9.6 ±20.7 [-3.6/22.7] p = 0.225 | 39.9 ±11.3 [32.8/47.1] | +21.9 ±15.0 [12.4/31.4] **p = 0.001**\*\*\* | +9.8 ±21.6 [-3.9/23.5] p = 0.147 | 41.0 ±15.3 [31.3/50.8] | +10.8 ±21.4 [-2.8/24.4] p = 0.114 |
| Control group | -6.5±21.1 [-20.7/7.7] p = 0.326 | 20.3 ±6.1 [16.4/24.2] | -6.5 ±21.1 [-20.7/7.7] p = 0.334 | -4.9 ±20.8 [-18.9/9.1] p = 0.461 | 23.1 ±10.8 [16.2/29.9] | -4.9 ±20.8 [-18.9/9.1] p = 0.452 |
| | H(4) = 8.593 p = 0.072 | | H(4) = 16.599 **p = 0.002**\*\* | H(4) = 16.494 **p = 0.002**\*\* | | H(4) = 7.635 p = 0.106 |

M = arithmetic mean; SD = standard deviation; CI = confidence interval of the mean; p = statistical significance; p (Δ >0) one sample *t*-test (bootstrapped, bias-corrected, and accelerated, 10.000 samples) to test significant differences of change Δ from 0; H (df) = test statistic (degrees of freedom) of Kruskal-Wallis H test, comparing changes Δ between groups

| After IPR + polishing | After 24h of demineralization | After 18 days of cyclic de- and remineralization |

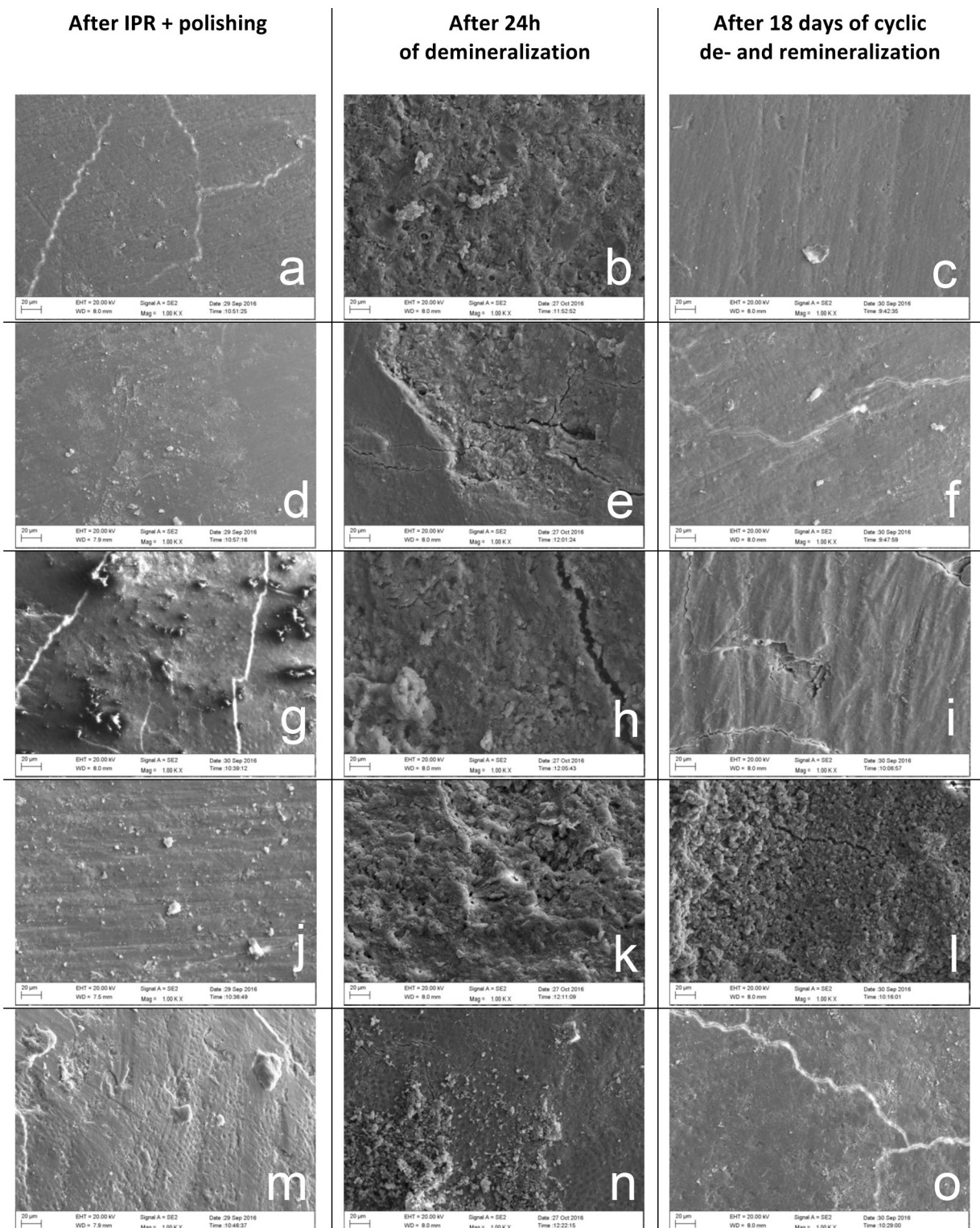

**Fig 2. Scanning electron microscopic images of enamel surfaces of randomly selected specimens of each experimental group.** (a) GMS after IPR, (b) GMS after demineralization, (c) GMS, (d) KAS after IPR, (e) KAS after demineralization, (f) KAS after demineralization and remineralization, (g) IOS after IPR, (h) IOS after demineralization, (i) IOS after demineralization and remineralization, (j) STB after IPR, (k) STB after demineralization, (l) STB after demineralization and remineralization, (m) control group untreated, (n) control group after demineralization, (o) control group after demineralization and remineralization.

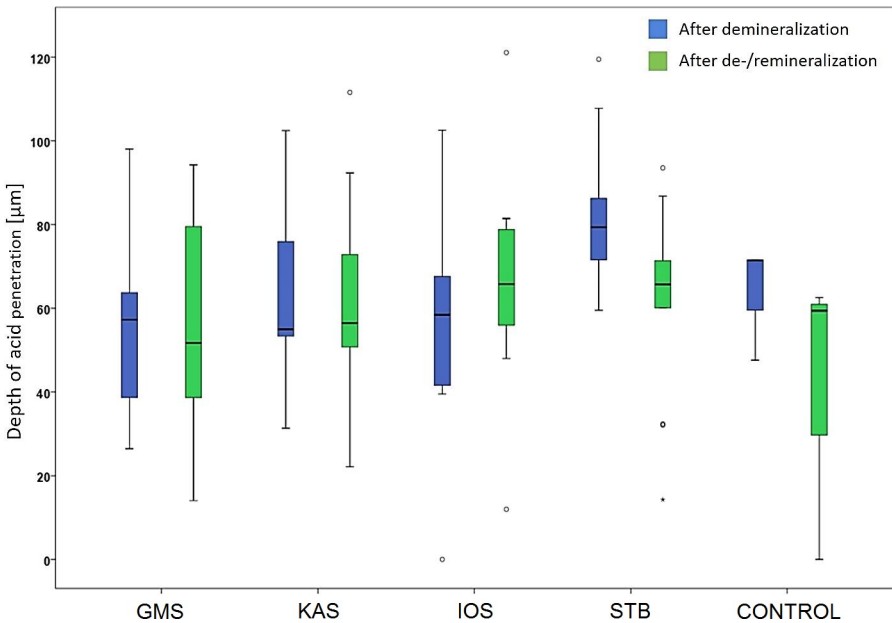

**Fig 3. Depth of acid penetration into interproximal enamel after 24 h of demineralization.** Thirty-seven degrees Celsius, mesial tooth surfaces and interchanging 24 h cycles of demineralization and remineralization for 18 days (distal tooth surfaces) after IPR treatment with the respective IPR systems and untreated controls.

### Polarization microscopy

The average depth of acid penetration after the 24 h demineralization treatment amounts to 57.2–75.9 µm and, after 18 days of cyclic demineralization and remineralization, to 50.9–76.6 µm (Figs 3 and 4). Conducting an IPR or the type of IPR system used had no significant effect on acid penetration depth into interproximal enamel after demineralization: H (4) = 8.360; p = 0.079. Significant differences were observed after cyclic demineralization and remineralization, as both the enamel surfaces treated with KAS and STB showed a significantly higher acid penetration depth: H(4) = 9.935; p = 0.042.

Enamel surface roughness after IPR did not correlate significantly with acid penetration depth after either subsequent demineralization (24 h, p = 0.151, p = 0.284) or cyclic demineralization and remineralization (18 days, p = 0.061, p = 0.67). In addition, no significant correlation was found between surface roughness and acid penetration depth after demineralization (24 h, p = –0.154, p = 0.29), whereas after cyclic demineralization and remineralization, enamel surface roughness correlated significantly, but to a minor degree, with acid penetration depth (p = 0.286, p = 0.047).

### Discussion

Rough enamel surfaces cause increased plaque formation [5–11]. Smooth surfaces reduce bacterial adhesion [12,13]. It is clinically established that interproximal enamel reduction (IPR) followed by polishing does not cause increased enamel roughness and that the genesis of caries requires the production of acids demineralizing enamel as the main factor for tooth decay [14–16].

The profilometry evaluation of enamel surface roughness revealed that after polishing surfaces that had been treated with IPR systems, these possessed a significantly greater smoothness than untreated enamel. These results are in accordance with other studies [17–19]. Teeth

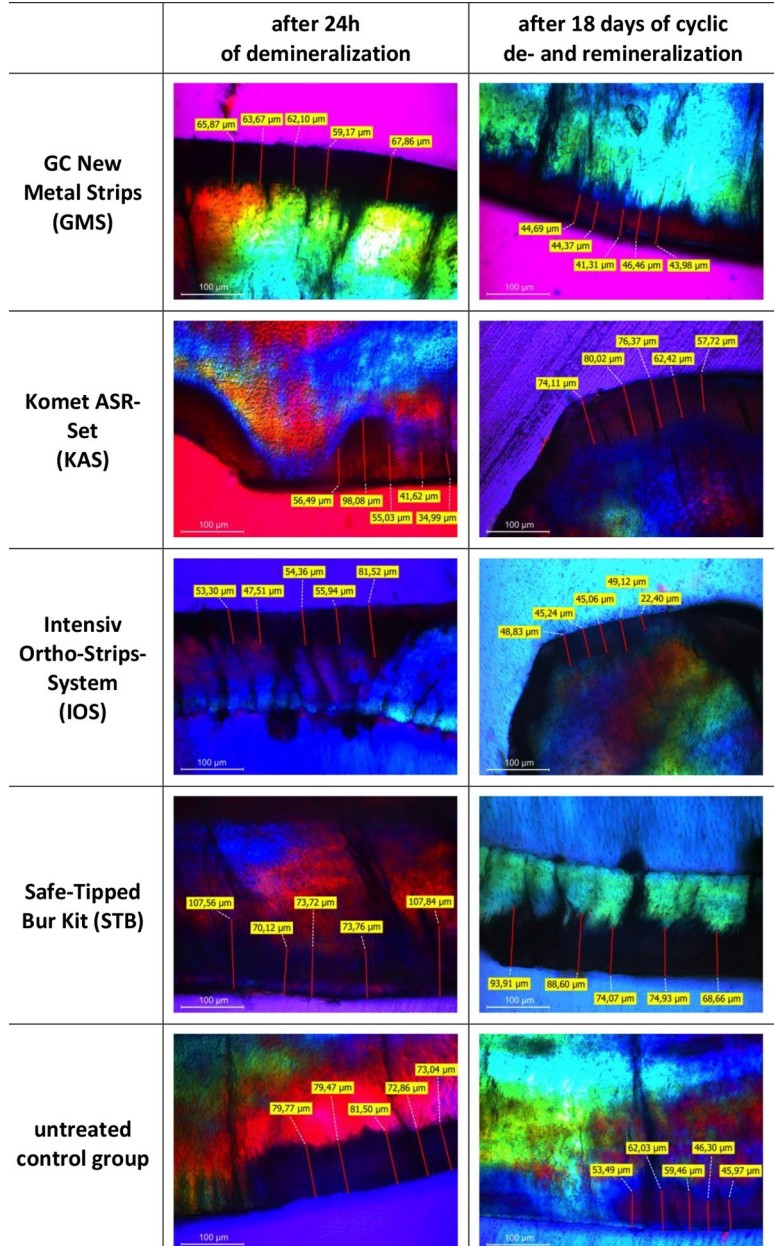

**Fig 4. Polarization microscope images showing acid penetration into interproximal enamel of randomly selected specimens of each experimental group.** (a) GMS after demineralization, (b) GMS after demineralization and remineralization, (c) KAS after demineralization, (d) KAS after demineralization and remineralization, (e) IOS after demineralization, (f) IOS after demineralization and remineralization, (g) STB after demineralization, (h) STB after demineralization and remineralization, (i) control group after demineralization, (j) control group after demineralization and remineralization.

treated by GMS, KAS, and IOS systems showed a significant reduction of surface roughness, whereas this was not the case for the STB system, which produced higher roughness values after polishing than before IPR, which coincides with the divergent statements by Arman et al. [20], Danesh et al. [17], and Radlanski et al. [5]. The profilometry as well as the SEM results revealed that the polishing effect and the influence of physiological remineralization of dental

tissues are important to avoid plaque retention. The present study also reveals some limitations. Factors such as missing bacterial influence and the age of the examined teeth could not be considered in this study [21].

Because of its high resolution, SEM is a good procedure to evaluate images visually [18]. Demineralized surfaces in this study showed height differences of the enamel, which represented a rough surface. When looking at the extent of demineralization after 24 h, all four tested IPR systems produced an increase in enamel surface roughness by demineralization despite the polishing performed after IPR. Interestingly, compared to the initial enamel roughness before IPR, the GMS and IOS systems yielded comparable roughness values after demineralization, indicating that these systems have an advantage in this respect over the STB and KAS systems, presumably because they were able to smooth the enamel surface in combination with the polishing performed to such an extent that the increase of roughness during demineralization simply restored the original roughness of the interproximal enamel.

These effects were even more evident after 18 days of cyclic demineralization and remineralization. Whereas an increase in surface roughness was also evident after 18 days in nearly all experimental groups treated with IPR and polishing, this was not the case for the IOS system. Interproximal enamel surfaces treated by the GMS and IOS systems and polishing had either comparable or even smoother enamel surfaces in relation to the initial surface roughness before IPR and were also associated with significantly lower depths of acid penetration than the KAS and STB systems were. Finally, our results show a significant correlation between surface roughness after demineralization and remineralization cycles for 18 days and acid penetration depth. This would indicate that polishing and remineralization smoothing of the enamel surface after IPR could possibly reduce the risk of enamel demineralization and interproximal caries. This concurs with results from previous studies that reported that IPR does not increase the risk of iatrogenic tooth decay [7]. In clinical orthodontics, the manual IPR technique (GMS) as well as the oscillating (IOS) system may be of advantage when deciding on IPR therapy, considering future caries susceptibility of the IPR-treated surfaces. In this regard, one other fact to consider clinically is the efficacy of IPR with different systems, which has been shown to be superior with oscillating systems [22], whereas diamond-coated manual strips have been shown to be associated with a rapid loss of abrasive power during use [23].

## Conclusion

Within the limitations of this study, the following conclusions can be drawn:

- Enamel surface roughness after 18 days of interchanging 24 h demineralization and remineralization following interproximal enamel reduction (IPR) and polishing correlated significantly with acid penetration depth into interproximal enamel.

- IPR-treated and polished enamel in general showed a reduced surface roughness compared to untreated enamel before IPR. Proper polishing of IPR-treated surfaces is thus advisable irrespective of the IPR procedure used, to minimize caries susceptibility.

- Remineralization processes reduced the depth of acid penetration in the untreated as well as the IPR-treated groups. Proper remineralization procedures after IPR are thus important, irrespective of the IPR method used, to minimize caries susceptibility.

- The manual IPR method using GMS and even more the oscillating IPR system (IOS) yielded smoother interproximal enamel surfaces and less acid penetration depth than the KAS and STB IPR systems after polishing and 18 days of cyclic demineralization and remineralization.

The GMS and IOS systems might thus have a clinical advantage in reducing caries susceptibility after IPR.

## Supporting information

**S1 Data. Raw data file.**
(XLSX)

## Acknowledgments

We thank Susanne Haussmann for technical laboratory assistance.

## Author Contributions

**Conceptualization:** Gholamreza Danesh, Wolfgang H. Arnold.

**Formal analysis:** Christian Kirschneck.

**Investigation:** Pascal Kai Konstantin Podstawa.

**Methodology:** Wolfgang H. Arnold.

**Supervision:** Gholamreza Danesh, Wolfgang H. Arnold.

**Visualization:** Gholamreza Danesh.

**Writing – original draft:** Gholamreza Danesh, Pascal Kai Konstantin Podstawa, Cate-Emilia Schwartz.

**Writing – review & editing:** Cate-Emilia Schwartz, Christian Kirschneck, Mozhgan Bizhang, Wolfgang H. Arnold.

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
