## [Decision Letter · Decision Letter 0]

19 Dec 2019

PONE-D-19-25991

Depth of acid penetration and enamel surface roughness associated with different methods of interproximal enamel reduction

PLOS ONE

Dear Prof. Dr. Danesh,

Thank you for submitting your manuscript to PLOS ONE. After careful consideration, we feel that it has merit but does not fully meet PLOS ONE’s publication criteria as it currently stands. Therefore, we invite you to submit a revised version of the manuscript that addresses the points raised during the review process.

We would appreciate receiving your revised manuscript by Feb 02 2020 11:59PM. To enhance the reproducibility of your results, we recommend that if applicable you deposit your laboratory protocols in protocols.io, where a protocol can be assigned its own identifier (DOI) such that it can be cited independently in the future. For instructions see: http://journals.plos.org/plosone/s/submission-guidelines#loc-laboratory-protocols

We look forward to receiving your revised manuscript.

Kind regards,

Thiago Saads Carvalho, Privatdozent, PhD

Academic Editor

PLOS ONE

Additional Editor Comments:

This is an interesting study, and I would also like to offer possible points of improvement:

1. The authors performed profilometry on the proximal surface, but the teeth had been mounted on silicon (with contact points). If there were contact points, how was profilometry performed? Were the teeth removed from the silicon and afterwards replaced to continue the experiment? Could you clarify this issue on the paper?

2. The authors use codes for the systems, which are presented in table 1, but Tables 2 and 3 do not contain the same codes. For examples, KAS is not present. Standardizing these codes would help the readers. Also, the same codes should be used in Figure 3, to allow the readers to immediately identify the systems.

3. Table 2 shows roughness values initially for both mesial and distal surfaces. Is this an average of both surfaces? However, Table 3 shows roughness after 24h demin(only for mesial surfaces) and after 18 days cycle (only for distal). Why were both surfaces not measured at both times, like it was doen in table 2? If they were really measured separately, differences between the two surfaces could compromise the analyses.

Please also take notice of other comments made by the other reviewer, presented below.

Journal Requirements:

2. In keeping with usual publication standards, please remove any copyright and trademark symbols from the manuscript text.

"We have no financial interest in this study."

Please provide an amended Funding Statement that declares *all* the funding or sources of support received during this specific study (whether external or internal to your organization) as detailed online in our guide for authors at http://journals.plos.org/plosone/s/submit-now Please state what role the funders took in the study.  If any authors received a salary from any of your funders, please state which authors and which funder. If the funders had no role, please state: "The funders had no role in study design, data collection and analysis, decision to publish, or preparation of the manuscript."

Reviewers' comments:

Reviewer's Responses to Questions

**Comments to the Author**

1. Is the manuscript technically sound, and do the data support the conclusions?

Reviewer #1: Yes

2. Has the statistical analysis been performed appropriately and rigorously? 

Reviewer #1: Yes

3. Have the authors made all data underlying the findings in their manuscript fully available?

Reviewer #1: Yes

4. Is the manuscript presented in an intelligible fashion and written in standard English?

Reviewer #1: No

5. Review Comments to the Author

Reviewer #1: Thank you for the opportunity to revise the paper “Depth of acid penetration and enamel surface roughness associated with different methods of interproximal enamel reduction”. The article analyzed the enamel surface quality comparing different IPR procedures, and it focused on the importance of remineralization to improve enamel surface quality. The paper is very interesting since IPR techniques have been gaining popularity in last years especially with new clear appliances’ treatments. There are some minor flaws to correct to improve the contents.

- First of all, the paper should be edit by native English because the language is inadequate and needs to be improved.

- Abstract, Conclusion section: the authors should explain the importance of remineralization after IPR procedures

- Considering the lack of literature on the topic, I suggest to add the following recent papers:

1. Comparison of the abrasive properties of two different systems for interproximal enamel reduction: Oscillating versus manual strips. Gazzani, F., Lione, R., Pavoni, C., Mampieri, G., Cozza, P. 2019. BMC Oral Health. 19(1),247

2. In vitro and in vivo evaluation of diamond-coated strips. Lione, R., Gazzani, F., Pavoni, C.,, Tagliaferri, V., Cozza, P. 2017. Angle Orthodontist. 87(3), pp. 455-459

- Conclusion section: the authors should give some clinical implications of their findings and highlight the clinical importance to use remineralization after IPR procedures.

6. PLOS authors have the option to publish the peer review history of their article (what does this mean?). If published, this will include your full peer review and any attached files.

Reviewer #1: No

---

## [Author Response · Author response to Decision Letter 0]

3 Feb 2020

Response to the reviewers’ comments

Additional Editor Comments:

1. The authors performed profilometry on the proximal surface, but the teeth had been mounted on silicon (with contact points). If there were contact points, how was profilometry performed? Were the teeth removed from the silicon and afterwards replaced to continue the experiment? Could you clarify this issue on the paper?

Response: As correctly assumed teeth were demounted from the mounting silicon mold for profilometry analysis and then reinserted in their original position to continue the experiment. We added this information to the materials and methods section of the paper.

2. The authors use codes for the systems, which are presented in table 1, but Tables 2 and 3 do not contain the same codes. For examples, KAS is not present. Standardizing these codes would help the readers. Also, the same codes should be used in Figure 3, to allow the readers to immediately identify the systems.

Response: Thank you for bringing this to our attention! We standardized the abbreviations accordingly in all tables, Figure 3 and the manuscript.

3. Table 2 shows roughness values initially for both mesial and distal surfaces. Is this an average of both surfaces? However, Table 3 shows roughness after 24h demin (only for mesial surfaces) and after 18 days cycle (only for distal). Why were both surfaces not measured at both times, like it was done in table 2? If they were really measured separately, differences between the two surfaces could compromise the analyses.

Response: Indeed the initial roughness values are an average of mesial and distal surfaces, as these were pooled for descriptive statistics as stated in the legend of Table 2, since roughness values between mesial and distal surfaces of teeth were quite similar at baseline and did not show significant differences in surface roughness. As both mesial and distal surfaces received identical IPR treatment, they were again both reevaluated after IPR with results shown in Table 2. Again we found no significant differences in surface roughness between mesial and distal surfaces after IPR treatment, which is why this was also reported as pooled results for mesial and distal surfaces in Table 2. From this point onward, mesial and distal surfaces received different treatments (24h demin or cyclic demin/remin for 18days respectively) forming two separate experimental groups, we decided to measure the clinically relevant endpoints rather than identical timepoints for both surfaces = experimental groups. As after IPR no differences were found for enamel roughness, there should be no compromise to the analysis, as the endpoints reported in Table 3 show, how in particular the two different treatments (24h demin or 18 cyclic demin/remin respectively) affected the IPR-treated enamel surface, irrespective of the mesial/distal aspect. The mesial and distal surfaces of each tooth were only used to be able to have the same enamel quality available for both experimental groups (24h demin or 18h cyclic demin/remin) thus minimizing biasing effects on the results (“split-tooth” model), which would have been more pronounced, if we had used double the number of teeth and only evaluated mesial surfaces. Measuring both surfaces at both times would have had no additional scientific or clinical gain, as mesial surfaces were covered in wax after 24h to protect them from further demineralisation, thus surface roughness of these mesial surfaces would have been the same as after 24h. The same is true for distal surfaces, which would most likely have had the same surface roughness after 24h of demineralisation as the mesial surfaces, as they started from the same baseline roughness, thus no additional clinically relevant information would have been gained.

Reviewer #1: 

1. First of all, the paper should be edit by native English because the language is inadequate and needs to be improved.

Response: The paper was revised by a professional English language service and the respective translation certificate is provided.

2. Abstract, Conclusion section: the authors should explain the importance of remineralization after IPR procedures.

Response: We added a respective explanation both to the abstract and the conclusions section.

3. Considering the lack of literature on the topic, I suggest to add the following recent papers: 

• Comparison of the abrasive properties of two different systems for interproximal enamel reduction: Oscillating versus manual strips. Gazzani, F., Lione, R., Pavoni, C., Mampieri, G., Cozza, P. 2019. BMC Oral Health. 19(1),247

• In vitro and in vivo evaluation of diamond-coated strips. Lione, R., Gazzani, F., Pavoni, C.,, Tagliaferri, V., Cozza, P. 2017. Angle Orthodontist. 87(3), pp. 455-459

Response: We added the suggested papers as suggested and briefly discussed their results.

4. Conclusion section: the authors should give some clinical implications of their findings and highlight the clinical importance to use remineralization after IPR procedures.

Response: We added a respective explanation and clinical implications to the conclusions section.

---

## [Editor Report · Decision Letter 1]

11 Feb 2020

Depth of acid penetration and enamel surface roughness associated with different methods of interproximal enamel reduction.

PONE-D-19-25991R1

Dear Dr. Danesh,

We are pleased to inform you that your manuscript has been judged scientifically suitable for publication and will be formally accepted for publication once it complies with all outstanding technical requirements.

With kind regards,

Thiago Saads Carvalho, Privatdozent, PhD

Academic Editor

PLOS ONE
---

## [Editor Report · Acceptance letter]

13 Feb 2020

PONE-D-19-25991R1 

Depth of acid penetration and enamel surface roughness associated with different methods of interproximal enamel reduction 

Dear Dr. Danesh:

I am pleased to inform you that your manuscript has been deemed suitable for publication in PLOS ONE. Congratulations! Your manuscript is now with our production department. 

With kind regards,

on behalf of

Dr. Thiago Saads Carvalho 

Academic Editor

PLOS ONE